

**Modelling confluence dynamics in large sand-bed braided rivers**
Haiyan Yang[1], Zhenhuan Liu[2]
[1]College of Water Conservancy and Civil Engineering, South China Agricultural
University, Guangzhou 510642, China; haiyan.yang2@gmail.com
[2]Guangdong Provincial Key Laboratory of Urbanization and Geo-simulation, School
of Geography and Planning, Sun Yat-sen University, Guangzhou 510275, China
Correspondence: liuzhh39@mail.sysu.edu.cn
**Abstract**
Confluences are key morphological nodes in braided rivers where flow converges,
creating complex flow patterns and rapid bed deformation. Field survey and laboratory
experimental studies have been carried out to investigate the morphodynamic features
in individual confluences, but few have investigated the evolution process of
confluences in large braided rivers. In the current study a physics-based numerical
model was applied to simulate a large lowland braided river dominated by suspended
sediment transport, and analyzed the morphologic changes at confluences and their
controlling factors. It was found that the confluences in large braided rivers exhibit
some dynamic processes and geometric characteristics that are similar to those observed
in individual confluences arising from two tributaries. However, they also show some
unique characteristics that are result from the influence of the overall braided pattern
and especially of neighboring upstream channels.
**Key words**: braided river, numerical model, confluence, dynamics, geometry, scour
hole



## 1. Introduction

Braided rivers are highly dynamic systems characterized by multiple frequently joining and bifurcating channels that form confluences and bifurcations. In these rivers, channel confluences and bifurcations are key morphological features whose dynamics and mutual interactions control many aspects of channel morphology and processes in braided networks. Exploring the mechanisms underlying confluence evolution is fundamental to better understand morphodynamic processes in braided rivers (Surian, 2015). Confluences as the junctions of two river branches have been widely studied both in field (e.g., Rhoads et al. 2009; Riley et al., 2015) and in laboratory (e.g., Ribeiro et al., 2012; Guillén-Ludeña et al., 2016), with a few focusing on large-scale confluences (e.g., Szupiany et al., 2009; Gualtieri et al., 2018). However, confluences in large braided rivers have rarely been investigated, mainly due to the lack of adequate methodologies to investigate large rivers with frequent channel migrations. The evolution and morphology of confluences in large braided rivers share some common features with junctions of two branches in a non-braided river. However, they might also be affected by the evolution of the overall braided pattern, especially by morphologic changes in their immediate upstream neighbourhood, thereby exhibiting unique morphodynamic processes and characteristics.

Channel confluences are important sites where adjustments in flow structure, sediment transport and channel morphology occur to accommodate convergence of water and sediment from different branches (Ferguson et al., 1992; Rhoads et al., 2009). Common morphologic features often include a scour hole typically oriented along the




direction of maximum velocity, avalanche faces at the mouth of each branch, sediment
deposition within the stagnation zone at the upstream junction corner, and bars formed
within the flow separation zone (Rhoads and Kenworthy, 1995, 1998; Best and Rhoads,
2008). The main factors that control flow structure and channel morphology at
confluences include flow and sediment discharge ratios between the two confluent
channels, the confluence angle and its planform asymmetry, and the degree of bed
concordance between the confluent channels (e.g. Szupiany et al., 2009). Recent
findings indicate that bifurcation asymmetry is not solely controlled by flow discharge
but is rather the result of multiple factors, including varying flow discharge, changes in
bed morphology and cross-stream water surface slopes (Gualtieri et al., 2018). Guillén-
Ludeña et al. (2016) found that the abundant sediment load of the dominant branch
plays a major role in controlling the dynamics of mountain river confluences.
The center of a confluence often features a scour hole with considerable erosion
depth. The scour zone may change from trough-shaped to more basin-like as the
confluence angle increases (Ashmore and Parker, 1983; Best, 1986). The scour depth
of a confluence has been related to the confluence angle and the relative discharge of
the confluent channels (e.g., Best, 1988), which typically ranges from two to four times
the incoming branch channel depths, suggesting some scale invariance in junction
morphology (Parsons et al., 2008). The slopes of beds dipping into scour holes in large
braided rivers are often gentle, e.g., less than 5% in the Brahamaputra River (Best and
Ashworth, 1997).
Numerical models are useful tools to assist field research because they can provide



large datasets with sufficient spatial and temporal resolution to investigate river
morphodynamics. In recent years, numerical models based on the basic flow and
sediment transport equations, such as the depth-integrated Delft 3-D (Schuurman et al.,
2013; Schuurman and Kleinhans, 2015), HSTAR (Nicholas, 2013) and other models
(e.g., Jang and Shimizu, 2005a, b; Yang, 2013) have been developed and applied to
simulate braided rivers. These models provided new insights into the dynamic
processes of braided rivers and enriched theories for these systems. Yang et al. (2015,
2018) developed a 2-D physics-based model that divides sediment into multiple
fractions and riverbed into several vertical layers, and simulated rivers with
morphodynamics compared well with natural braided rivers. They analysed the
dynamic processes and statistical features in these rivers and investigated the key
factors controlling channel generation and disappearance.
The present paper applies that model (Yang et al., 2015, 2018) to simulate large
lowland sand-bed braided reaches. The main objectives of the study are to
quantitatively analyse changes in flow field, sediment concentration and bed elevation
at typical confluences, compare them with those observed in natural rivers, and
investigate evolution processes at confluences and the controlling factors. Results of
this study will expand the current knowledge on confluence dynamics in large sand-bed
braided rivers and provide the opportunity to analyse similarities and differences
between braided rivers and other river types.



**2. Model Descriptions and Methods**
**2.1 Numerical model and solutions**
A two-dimensional depth-integrated numerical model was applied to simulate the
confluence dynamics in the lower reaches of large sand-bed braided rivers. The
hydrodynamic model consists of a mass and two momentum conservation equations,
which are derived from the three-dimensional Reynolds equations for incompressible
and unsteady turbulent flows by depth integrated. The hydrodynamic equations are
solved using the Alternating Direction Implicit method, which has been widely used in
the solution of shallow water equations (e.g. Lin and Falconer, 2006).
The sediment transport is described by a two-dimensional solute transport
equation and a bed deformation equation, with a fractional method adopted to simulate
the sorting process of graded sediments. By dividing the graded sediments into $N$
fractions, the transport of the $k$th size fraction is calculated by

$$\frac{\partial Hs_k}{\partial t} + \frac{\partial HUs_k}{\partial x} + \frac{\partial HVs_k}{\partial y} = -\alpha_k \omega_k (s_k - \phi_k) + \frac{\partial}{\partial x}\left(D_{xx}H\frac{\partial s_k}{\partial x} + D_{xy}H\frac{\partial s_k}{\partial y}\right)$$
$$+ \frac{\partial}{\partial y}\left(D_{yx}H\frac{\partial s_k}{\partial x} + D_{yy}H\frac{\partial s_k}{\partial y}\right) \tag{4}$$

The riverbed deformation is given as

$$(1-p_0)\frac{\partial z_b}{\partial t} = \frac{1}{\gamma_s}\sum \alpha_k \omega_k (s_k - \phi_k) \tag{5}$$

where $s_k$ is the sediment concentration for size fraction $k$; $\alpha_k$ is the adjustment coefficient
for size fraction $k$; $\omega_k$ is the fall velocity for size fraction $k$, calculated by the equation
of van Rijn (1984); $\phi_k$ is the transport capacity for size fraction $k$; $D_{xx}$, $D_{xy}$, $D_{yx}$ and $D_{yy}$





are depth-averaged dispersion–diffusion coefficients in the $x$ and $y$ directions,
respectively, with Preston's (1985) equations adopted; $p_0$ is the porosity of bed layer
sediments; $z_b$ is the riverbed elevation; and $\gamma_s$ is the specific weight of sediment.
In order to account for the influence of bed composition, a multiple layer method
is used to simulate the spatial and temporal variations of sediment gradations in the
loose bed layers. The sediment transport equations were solved with the Ultimate
QUICKEST Scheme, which was developed to simulate 2-D solute and mass transport
with high concentration gradients (Lin and Falconer, 1997). More details on the model
and solution method can be found in Yang (2013), Yang et al (2015), Zhou et al (2003),
and Zhou and Lin (2006).
**2.2 Model settings and boundary conditions**
The model was set up based on the data collected from the lower reaches of
Jiahetan and Huayuankou in the Yellow River in China (Zhao et al. 1998; Wu 2007),
including flow discharge, bed slope, sediment size distribution and channel dimension.
The simulated river was approximately 50 km long and 5 km wide, initially straight and
plane, with a uniform bed slope of 0.000233. The model divided the sediments into six
fractions, with particle sizes ranging from 0.0025 to 0.25 mm. Two spur dikes were set
up at the right and left bank near the upstream boundary to increase the local flow speed,
aiming to accelerate the initial channel evolution process. The input flow discharge was
given as 6250 m³/s, and the sediment concentration was set to 44.5 kg/m³ referred to
the field data of the Yellow River.



**3. Channel Confluences**

**3.1 General processes**

Instabilities in the simulated braided river were initiated in the alternate shallow and deep areas near the upstream spurs. This induced flow disturbance and caused intense erosion and deposition in the neighbouring areas and subsequently downstream (Figure 1). A braided pattern was then formed through the development of multiple row bars, which is one of the two most common mechanisms of braided pattern evolution in rivers (Ashmore, 2009). Channels divided and rejoined around bars, forming nodes typical in braided rivers—confluences and bifurcations, usually with deep scour holes in their center.

*<Figure 1 insert here>*

Figure 1 Sequential evolution of confluences in the modelled river (water depth/m)

One confluence and its upstream bifurcation, with the two converging branches and their surrounding bar, form a pool-bar unit, the basic element of a braided river. The confluence of a pool-bar unit is simultaneously the bifurcation of another pool-bar unit and also represents the branch bend scouring pool of a third confluence. For example, confluence D is the scouring pool of the right branch of confluence E in Figure 1. Nodes can also evolve and change their roles. For example, the scouring pool at the outer bank of two branches of a confluence can travel downstream to renew the confluence. This is what happened to pool 1 (day 26 in Figure 1), which continued



extending downstream and merged with confluence C on day 33. Confluences can also
migrate downstream. High flow can cause fine sediment erosion at a bifurcation and at
the two front sides of its downstream bar head, with subsequent transport of the eroded
sediments downstream and deposition at the bar-tail confluence, thereby causing
infilling of the confluence head and scouring of the confluence tail. One example of
this mechanism is shown by confluence E. Downstream movement of a confluence is
also common in natural rivers and has been suggested to be controlled by aggradation
in the confluence area and local avulsions of the primary channel (Roy and Sinha, 2007).

9        The pool in the dominant branch of a confluence often developed at the front of

the channel bend and featured a substantial scouring depth (e.g., pool 1 in Figure 1).
Conversely, the pool in the secondary branch tended to develop behind the channel
bend and was characterized by a relatively shallow scouring depth (e.g., pool 2 in Figure
1). However, sometimes in a thin pool-bar unit formed in the early stage, the scouring
pools of both branches developed at the front of the channel bend. One example is pool-
bar unit 3 in Figure 1 on day 26. As the bar grew laterally and shortened, the pool
migrated downstream across the bend.
**3.2 Geometry and controlling factors**

18       Confluences are normally located in areas with deepest flow due to the fact that

where two or more branches meet intense erosion occurs, thereby removing a large
amount of sediments. Figure 2 shows the cross-sectional maximum erosion depth
compared to river geometry in a fully evolved braided river, with A–G indicating the
location of typical confluences both in the river and along the corresponding erosion

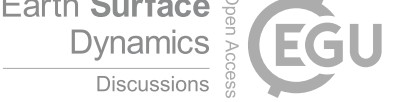



curve. The maximum erosion depth curve exhibits a periodic wave pattern with peaks
and valleys, representing local minimum and maximum erosion depths, respectively.
Generally, most of the cross-sectional topographic valleys are located at confluences,
while the remaining valleys are often scouring pools at channel bends between two
confluences. Confluence erosion tends to be more intense when the total confluence
channel width is narrower.
*<Figure 2 insert here>*
Figure 2 Cross-sectional maximum erosion depth compared to river geometry in a fully
evolved river (day 33)
Table 1 Parameters of seven typical confluences in a fully evolved river (day 33)
*<Table 1 insert here>*
Table 1 lists the gross hydraulic and geometric features of confluences A–G.  The
deepest scour hole mostly occurred at the confluences with two branches most similar
to each other. For example, despite not having the fastest flow or largest discharge,
confluence E (discharge ratio being 1.16, closest to 1) still developed the deepest scour
hole (4.01 m). Nevertheless, the confluence with branches least similar to each other
(e.g. discharge ratio being 3.27 for confluence A) also formed a remarkably deep scour
hole (3.99 m), when the dominant branch played a key role in this process.
The angle between the two branches of the simulated confluences increased with

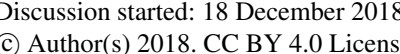

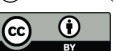

decreasing discharge ratio, while it was also influenced by the morphological changes
in surrounding areas, especially by upstream channel evolution. For example, with the
lowest discharge ratio, the two branches of confluence E developed a larger confluence
angle than most of the other confluences. However, confluence F had relatively high
discharge ratio but developed the largest branch angles, too. This might be due to the
fact that the flow direction of the secondary branch (left branch for confluence F) was
largely determined by the flow of its upstream bifurcation.
The confluence scour axis tends to parallel the dominant branch, which has been
observed in laboratory experiments (e.g. Ashmore and Parker, 1983; Best, 1987),
forming a smaller angle with it, with the exception of confluences C and G. On one
hand, faster flow existing in the dominant branch eroded more sediments from the
riverbed and formed the scour hole head. On the other hand, the flow direction of a
confluence oriented towards the dominant branch, determining the scour hole axis
direction. However, at confluences C and G, the confluence scour axis was directed
towards the secondary branch. For confluence C, the scour hole intruded into the
secondary (left) branch (Figure 3a), so that the hole direction was mainly determined
by the secondary branch flow. For confluence G, flow was influenced by upstream
channel evolution and was mostly parallel to the secondary (right) branch, resulting in
the scour hole axis direction oriented to the secondary branch as well.
**3.3 Morphology and evolution process**
Figure 3 shows the evolution process of confluences B–F. Generally, the evolution
trend of the overall braided pattern controlled the generation and disappearance of





confluences. In 15 days the ridge between the two branches of confluence B was eroded
away and the right branch grew to be the dominant one, with the main direction of flow
and scour hole at the confluence switching from the left to the right branch (Figure 3b).
Confluence C moved downstream to merge with its neighborhood pool (pool C') and
became the deepest confluence, with its downstream channel (channel 1) becoming the
largest channel in the river. Confluences D and F disappeared (Figure 3a), mainly due
to the blockage of one of their branches. Confluence E shrank accompanied by a new
confluence (confluence H) generation.
*<Figure 3 insert here>*
Figure 3 Evolution process of confluences B–F (erosion depth/m): (a) 3-D channel
morphology; (b) 2-D plane map with depth-averaged velocity (m/s)

14       In particular, the significant growth of channel 1, closely related to the

enlargement of confluence C, controlled the consequent evolution of downstream
confluences, including D, E, F and H. In 15 days, confluence D gradually disappeared
because the rapid growth of one of its branches —channel 1 promoted the blockage of
the other branch. Due to the large amount of flow diverted into channel 1, confluence
E and its branches experienced a decrease in flow, resulting in sediment deposition and
overall confluence weakening. This also contributed to blocking branch 2 of confluence
F, thereby leading to its ultimate disappearance. Meanwhile, as channel 1 grew, water
overflowed out of its downstream channel bend, leading to the formation of a new



channel (channel 3) and a new confluence (confluence H).
A remarkably steeper bed developed at the mouth of confluence branches, similar
to avalanche faces in small-scale confluences. When one branch was obviously
dominant, there was no visible avalanche face at its mouth, as shown by confluences B
and F (Figure 3a), whose discharge ratios were 2.97 and 2.68, respectively (Table 2).
Conversely, avalanche faces often existed in their secondary branch. However, when
one branch did not fully dominate over the other one, avalanche faces generally
occurred at both of the branch mouths. For example, at confluence E that exhibits two
relatively equivalent branches in terms of discharge, there are two visible avalanche
faces in front of the scour hole, with digging slopes being 1.624% for the left branch
and 1.154% for the right branch, which are 70- and 50-folds of the original bed slope,
respectively. Compared to small-scale confluences, the relatively gentle scour slopes
observed in this study agreed with the findings of Szupiany et al. (2009) and Best and
Ashworth (1997) in large sand-bed rivers.
A ridge sometimes developed in one branch of a confluence, which was often a
newly formed branch that bifurcated from a channel bend. This typically happened in
the early stage of confluence evolution. The ridge was initially located between two
flow channels and as the new branch evolved, it was eroded away. Confluence B and
the newly formed confluence H illustrate the process of ridge evolution, where the ridge
can be viewed as a type of avalanche face.



**4. Morphodynamic Processes at a Typical Confluence**
Confluence E was chosen to perform an in-depth analysis of morphodynamic
changes occurring at a typical confluence.
**4.1 Evolution process**
Figure 4 shows the evolution process of confluence E and its two branches.
Confluence E experienced a period of expansion and then contraction from day 25 to
37, during which the dominant channel switched from the left to the right branch. The
right branch began to lose its dominant role around day 32, as the left branch
progressively increased in terms of size and discharge. Geometric changes in both
branches through time illustrated that, the width of one channel declined along with the
reduction in its flow discharge (Figure 6). As mentioned before, confluence E was
ultimately largely filled with sediments due to flow recession as flow being diverted
away from its upstream channel, when the right branch grew to play a dominant role.
*<Figure 4 insert here>*
Figure 4 Evolution process of confluence E (erosion depth/m)
Flow velocities at seven channel cross-sections on day 32 are shown in Figure 5,
with five located on confluence E and two located on its two branches. Flow was more
averagely distributed in the left branch than the right one. At the head of the confluence,
section E3 exhibited two velocity cores, with a zone of lower velocity occurring in the
central area where the two flows combined. At the immediately downstream section E4,



flow concentrated and accelerated, with the two cores becoming indistinguishable. And
at the subsequent section E5 where water was deepest, flow became even stronger with
just one major core existing in the hole centre. Flow velocity at sections E4 and E5
peaked close to the right bank, promoting more sediments eroded away from the right
bank and thus developing a steeper bank slope than the left one. Although water was
deepest at section E5, which was approximately located in the center of the confluence,
the fastest flow occurred at section E6, which was located toward the end of the
confluence. A similar pattern was also often observed at the other six confluences
shown in Figure 3. This might explain the commonly observed downstream migration
of confluence scour holes, with deposition occurring at the hole heads due to upstream
sediment deposition and erosion occurring at the hole tails due to contracted fast flow
transporting sediments away. The flow in the two branches seemed to mingle faster
than natural rivers (e.g. Szupiany et al., 2009), which might result from the rapid
changing mixed bed layers in the model.
*<Figure 5 insert here>*
Figure 5 Distribution of depth-averaged flow velocities through confluence E on day
32 (erosion depth/m)
Sequential changes of flow discharge for the two branches converging at
confluence E are illustrated in Figure 6. The left branch flow experienced a slight
decrease and then steadily increased up to a maximum of 1092 m$^3$/s on day 33. This





increase appeared to result partly from the disappearance of a middle channel between
two adjacent bars enclosed by the left and right branches and partly from channel
widening (Figure 4, day 25 to 32). Then the discharge gradually decreased down to 790
m³/s till day 40. Meanwhile, discharge of the right branch increased up until day 26 due
to channel constraint and the disappearance of a small bifurcation. After that, the
sinuosity of the right branch bend increased, ultimately resulting in an avulsion, with
the newly formed channel diverting a large portion of flow and consequently leading to
a discharge decrease down to a minimum of 576 m³/s. Between days 32 and 33 the two
branches showed very similar discharge values. Before that the right branch was the
dominant branch, while after that the left branch became dominant.
*<Figure 6 insert here>*
Figure 6 Sequential changes of flow discharge for the two branches of confluence E
**4.2 Scour hole**

15       A rapid displacement of the scour hole from the left to the right bank occurred

(Figure 4), which was intricately linked to the evolution of the left and right branches.
Specifically, the location of maximum erosion depth gradually moved from the left
bank to the midchannel from day 25 to 32 when the discharge ratio between the two
branches approached 1, and then migrated progressively closer to the right bank when
the discharge ratio increased above 1. These movements of the scour hole in response
to evolution of the two incoming flows further corroborate our previous observation
that, confluence dynamics are largely controlled by upstream channel morphology and



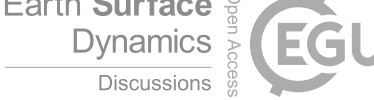

dynamics. Szupiany et al (2009) observed a similar process in field research, but they
suggested that the velocity has the most significant influence other than the discharge
ratio.

4        The orientation of the confluence angle was mainly controlled by the discharge of

its two branches. Initially, the discharge of the right branch was substantially larger than
that of the left branch (Figure 6) and the confluence axis aligned closely with the
direction of the right branch. The orientation angles of the two branches were 28° and
4°, respectively (Figure 7). As discharge decreased in the right branch and increased in
the left branch, the orientation angle of the right branch increased while the angle of the
left branch decreased. By day 32, the two branches had comparable discharges and the
scour axis approximately bisected the scour angle, with their orientation angles with
respect to confluence E equal to 30.5° and 29.5°, respectively. As found in the
experiment of Mosley (1976), the scour hole at confluence E enlarged and deepened
considerably (Figure 4, day 32). The bed morphology of the confluence is related to a
characteristic trough-shaped scour hole in the centre with a steeper front face than the
tail, which has been observed in laboratory flumes by Ribeiro et al (2012).
*< Figure 7 insert here>*
Figure 7 Changes in the orientation angle of the two branches of confluence E with
respect to the confluence axis
**4.3 Relationships between flow velocity, shear stress and bed elevation**

22       Simulation results for a cross-section on the left branch of confluence E (Figure 4,



day 28) were extracted to analyse factors influencing morphodynamic changes in flow
channel, with Figure 8 showing changes in flow velocity, shear stress and bed elevation
across the section over eight days.
*<Figure 8 insert here>*
Figure 8 Spatial distribution of flow velocity, water depth and shear stress across the
left branch of confluence E
Although the major peaks in flow velocity,  shear stress and flow thalweg were
initially located between the channel center and the right bank (peak 1 on day 15 in
Figure 8), their exact locations differed, with the peak in flow thalweg being closer to
the right bank. Over the next few days, shear stress and flow velocity continued to
decrease until day 20, thereby promoting sediment deposition and river bed becoming
shallower. The thalweg started to migrate by day 20 and was replaced by a newly grown
one near the left bank by day 23. On the contrary, the secondary peak in shear stress
close to the left bank (peak 2) continued to increase until it reached a value higher than
peak 1 by day 20. But during this period bed topography remained nearly unchanged.
As flow velocity increased in the left peak bank, shear stress reached its maximum
value across the channel and more sediments were eroded and removed from the
channel bed. Consequently, visible erosion occurred and river bed deepened near the
left bank (Figure 8, day 23), forming a new thalweg in the channel, with peaks in flow
velocity and shear stress occurring coincident to the thalweg. Generally, this process





indicates that increasing shear stress and flow velocity caused local erosion, resulting
in riverbed deepening. Importantly, there was a time lag before thalweg matched the
peaks.
**5. Conclusions**
In the present study, an existing numerical model was employed to simulate
natural large lowland braided rivers dominated by suspended sediment transport. The
morphodynamic processes and their controlling factors at confluences were
investigated and the following conclusions can be drawn.
1. In a braided river, a major change in the braiding pattern can affect the overall
evolution process of the confluences downstream, e.g. confluence generation and
enlargement, or decline and disappearance. Locally, flow from neighbouring upstream
channels often plays a key role in influencing the dynamics and geometry of a
confluence.
2. A steep bed slope similar to avalanche face in small-scale confluence can
develop at the mouth of the confluent branches, with its formation being related to the
degree of relative discharge dominance between the two branches. When one branch
has a fully dominating discharge, an avalanche face only occurs at the mouth of the
secondary branch; when the two branches have similar discharges, avalanche faces will
occur at the mouths of both branches.
3. The confluence scour hole is normally located close to the bank of the secondary
branch, which often has a steeper bank slope as the cross-sectional flow velocity peak
usually occurs close to the bank of the secondary branch. Downstream migration of a




scour hole is common due to sediment deposition at its head and erosion at its tail, with
maximum flow velocity occurring between the hole center and tail.
4. The discharge ratio between the two branches of a confluence controls its flow
direction, shape, depth and orientation, which is also influenced by the upstream flow.
As the discharge ratio decreases, the scour angle between the two branches enlarged
and the scour hole deepens. The confluence flow direction and scour axis usually tends
to be parallel to the dominant branch, and when the two branches become nearly
equivalent, the scour axis approximately bisects the scour angle.
5. Increased shear stress and flow velocity may cause local erosion and scour
deepening, when there is a time lag before the thalweg location coincides with the flow
peaks.
**Acknowledgements** We are grateful for the financial support received from the
Department of Science and Technology of the Guangdong Province (grant No.
2018A030310152), the National Natural Science Foundation of China (grant no.
41571172) and South China Agricultural University (grant No. 7600-218137).
**Compliance with Ethical Standards**.
**Conflict of Interest** None.

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

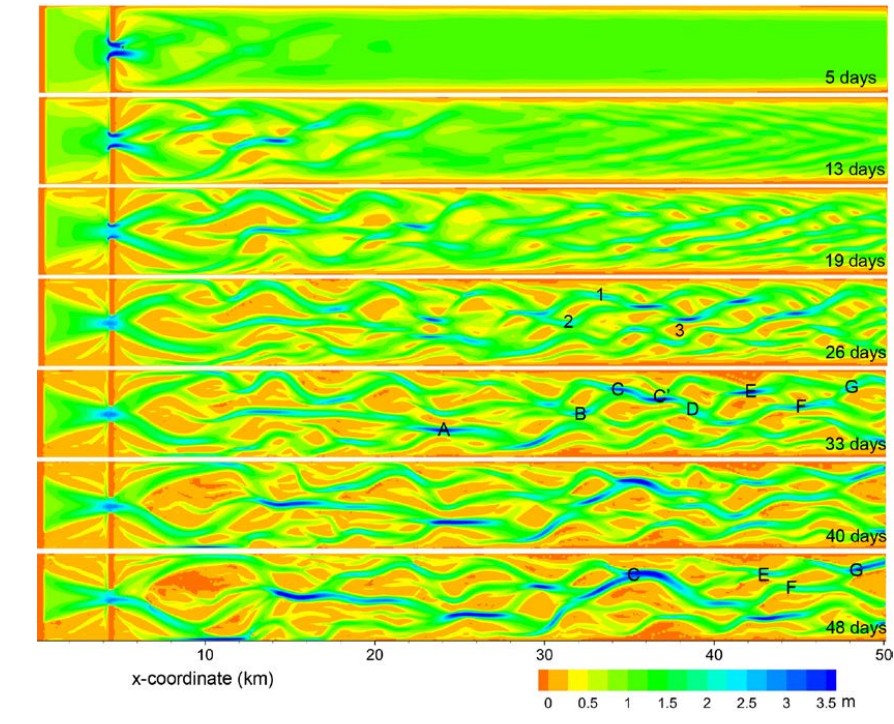

Figure 1 Sequential evolution of confluences in the modelled river (water depth/m)

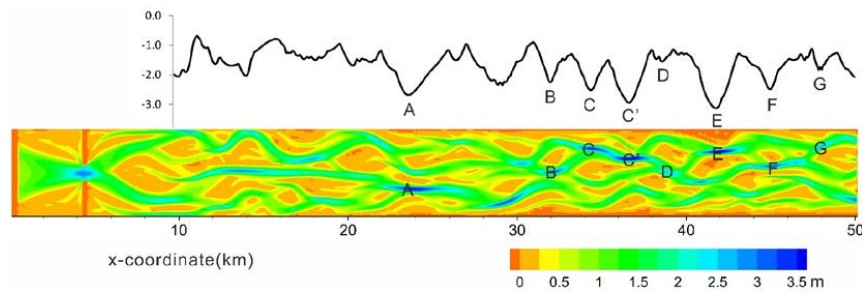

Figure 2 Cross-sectional maximum erosion depth compared to river geometry in a fully
evolved river (day 33)



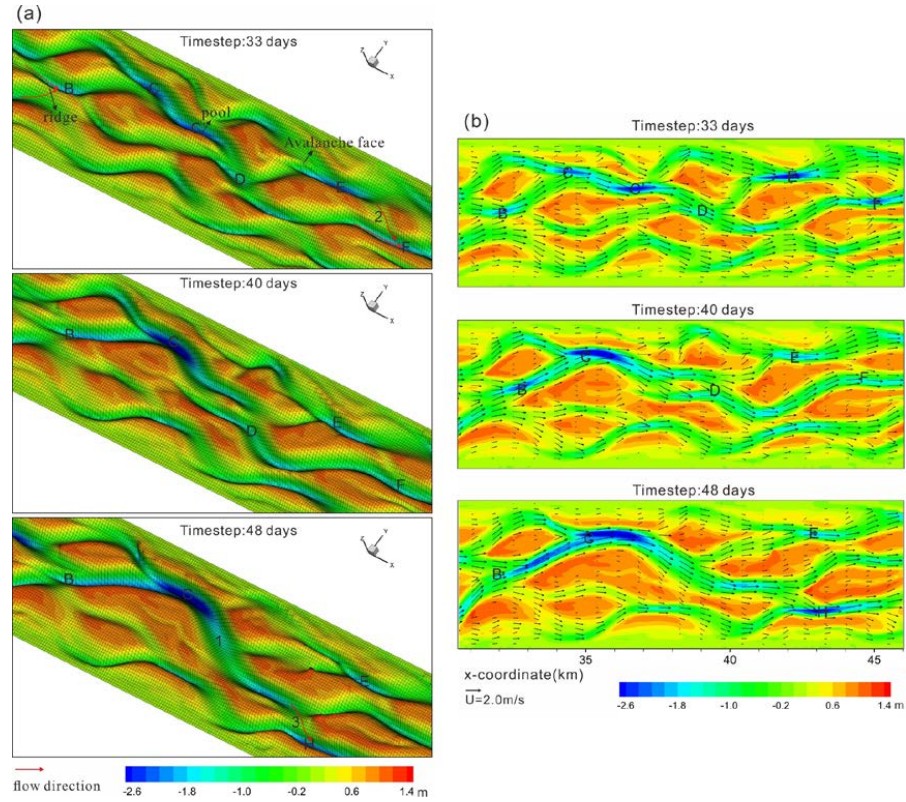



2  Figure 3 Evolution process of confluences B–F (erosion depth/m): (a) 3-D channel

3  morphology; (b) 2-D plane map with depth-averaged velocity (m/s)



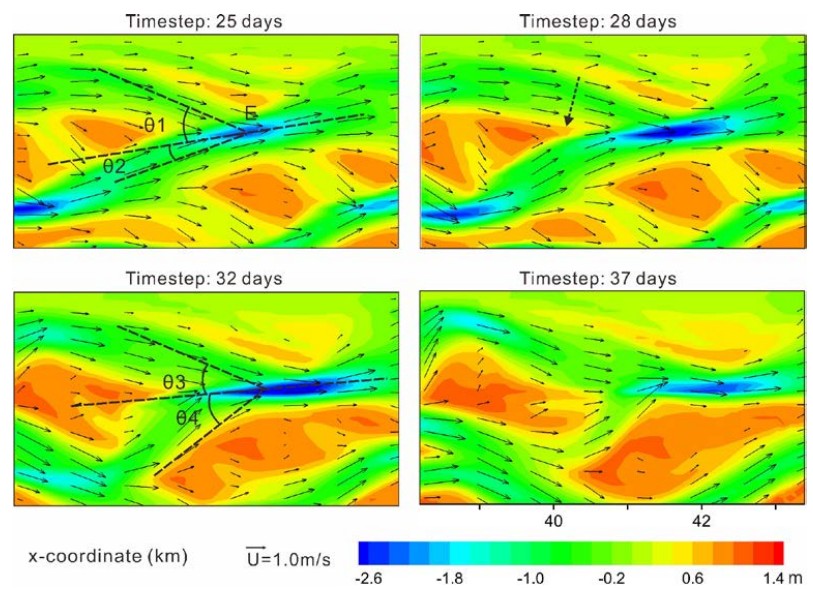

2    Figure 4 Evolution process of confluence E (erosion depth/m)

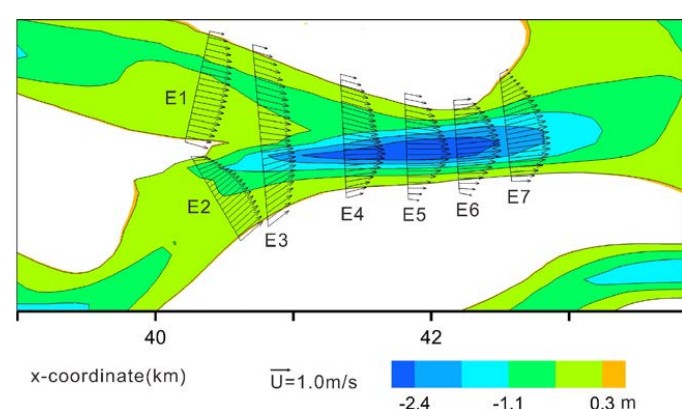

5    Figure 5 Distribution of depth-averaged flow velocities through confluence E on day

6    32 (erosion depth/m)



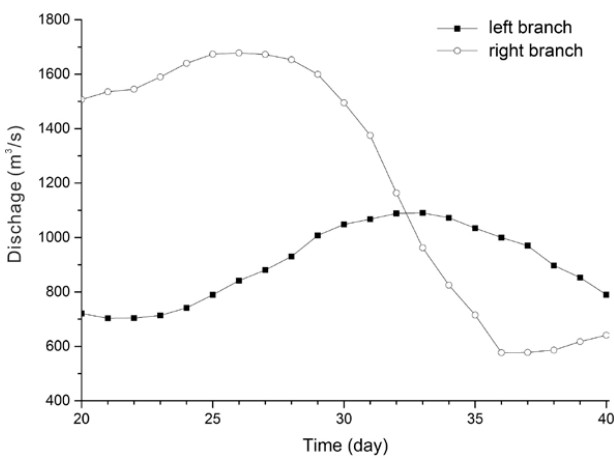

2    Figure 6 Sequential changes of flow discharge for the two branches of confluence E

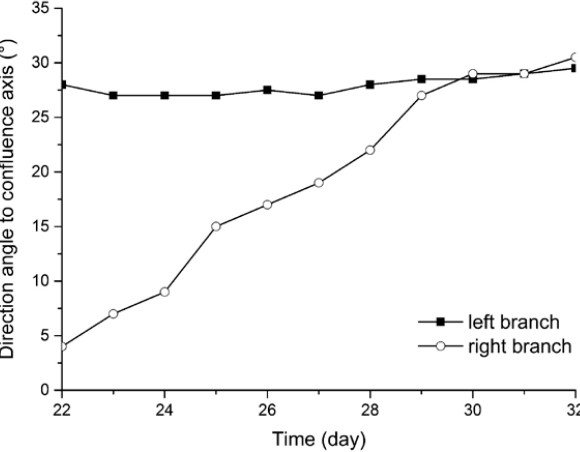

5    Figure 7 Changes in the orientation angle of the two branches of confluence E with

6    respect to the confluence axis



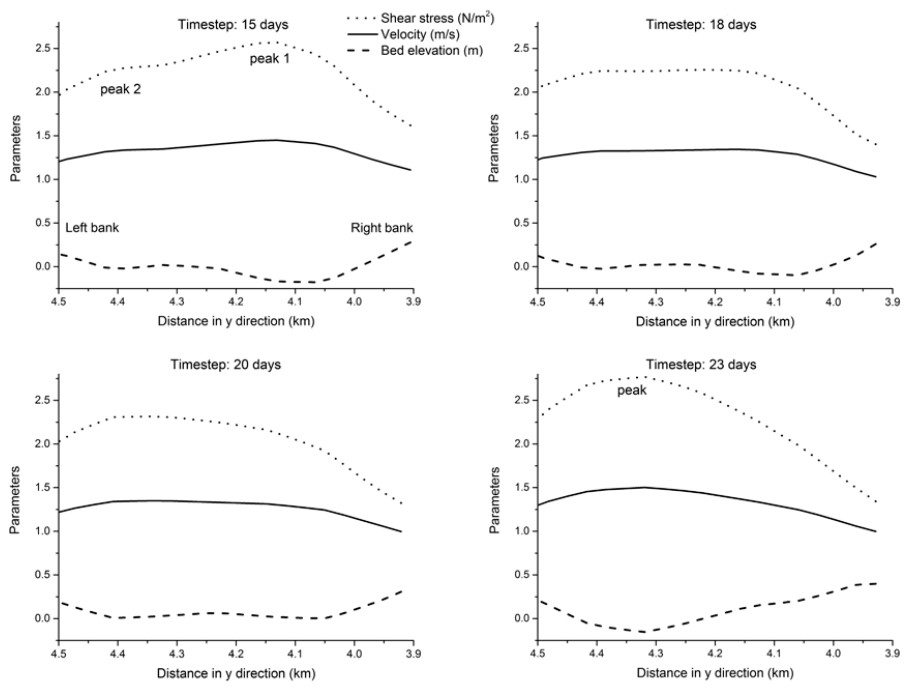

Figure 8 Spatial distribution of flow velocity, water depth and shear stress across the
left branch of confluence E
**Tables**

6        Table 1 Parameters of seven typical confluences in a fully evolved river (day 33)

| No. | Maximum scour depth (m) | Water depth (m) | maximum flow velocity (m/s) | Discharge of left branch (m³/s) | Discharge of right branch (m³/s) | Discharge ratio | Angle of two branches (°) | Angle to left branch (°) |
|-----|------|------|------|--------|--------|------|-------|-------|
| A   | -2.67 | 3.78 | 2.40 | 2133.1 | 652.4  | 3.27 | 34.22 | 8.10  |
| B   | -2.24 | 3.37 | 1.96 | 1599.3 | 539.3  | 2.97 | 34.68 | 15.55 |
| C   | -2.49 | 3.56 | 2.08 | 1385.0 | 1738.7 | 1.26 | 42.86 | 9.55  |
| D   | -1.51 | 2.58 | 1.73 | 1916.1 | 826.7  | 2.32 | 36.17 | 13.59 |
| E   | -3.10 | 3.99 | 2.12 | 1096.0 | 948.2  | 1.16 | 49.40 | 14.93 |
| F   | -2.46 | 3.42 | 2.10 | 498.3  | 1333.9 | 2.68 | 50.24 | 29.89 |
| G   | -2.31 | 3.27 | 2.08 | 1089.3 | 724.2  | 1.50 | 50.87 | 29.71 |

Note: Discharge ratio = discharge of dominant branch/discharge of secondary branch.