# Peer review of "Modelling confluence dynamics in large sand-bed braided rivers"

_Earth Surface Dynamics, 2018_

## Referee Comment (RC1) · Anonymous Referee #1 · 16 Jan 2019

The paper reports some qualitative aspects of anabranch confluence kinetics in braided rivers based on a computational morpho-dynamic model. This approach to describing and analyzing confluence dynamics is new and I am not aware of any similar published work. This topic has been approached previously using field observations and physical modeling. The paper contains 5 itemized conclusions. None of them are new and some have been well known in the literature for 20 years or more, including from some of the papers cited in the manuscript. For that reason alone I cannot support publication of the paper. However, the fact that the dynamics can be reproduced in a computational model is very useful, in which case the paper could be re-written to focus on the ways in which the model reproduces known aspects of confluence morphology and dynamics, and perhaps any observed differences for the sand bed case

versus existing gravel-bed examples.

**ESurfD**

Interactive
comment

---

## Referee Comment (RC2) · Anonymous Referee #2 · 4 Feb 2019

This manuscript describes the set-up and execution of a numerical model to simulate the morphodynamics of a sand-bed river. The model parameters are based upon a river sand-bed river. The model is run for 48 days of simulation time. The subsequent bed topography predictions are used to analyse confluence dynamics. Page 4 Line 14 states that "the main objectives of the study are to quantitatively analyse changes in flow field, sediment concentration and bed elevation at typical confluences, compare them with those observed in natural rivers, and investigate evolution processes at confluences and the controlling factors." Sections 3 and 4, which describe and discuss the results, are written in a fairly general style; the data presentation and analysis lacks the rigor that is necessary to investigate the manuscript's objectives. There is a dearth of quantitative comparison to natural rivers and the single simulation does not provide

the experimental framework that is necessary to answer the research question.

---

## Referee Comment (RC3) · Anonymous Referee #3 · 15 Feb 2019

The manuscript presents an investigation on confluence dynamics in braided rivers using a 2D model. The confluence evolution is one key process in braided rivers, and the 2D numerical model is a kind of useful tool to investigate the confluence dynamics. However, less novelty can be found regarding both modelling and mechanics of confluence process. For the model, in which six fractions are used for sediments with size from 0.0025 to 0.25mm, the interaction among fractions should be significant, but it is not clear how to deal with them and whether the cohesive is taken into account. For the results and discussion, outputs of simulation are just given directly with very limited contribution to the related knowledge.

---

## Author Comment (AC1) · 4 Mar 2019

Many thanks for the suggestions. Please see the supplement files for my responses, as equations in the file can not be pasted here.

Please also note the supplement to this comment:
https://www.earth-surf-dynam-discuss.net/esurf-2018-85/esurf-2018-85-AC1-supplement.pdf